# Bioactive Phenolic Compounds from Rambutan (*Nephelium lappaceum* L.) Shell: Encapsulation, Structural Stability, and Multifunctional Activities

**DOI:** 10.3390/ijms262210859

**Published:** 2025-11-09

**Authors:** Carlos Barba-Ostria, Orestes López, Alexis Debut, Arianna Mayorga-Ramos, Johana Zúñiga-Miranda, Elena Coyago-Cruz, Rebeca Gonzalez-Pastor, Kevin Cartuchi, Antonella Viteri, Ana Belén Peñaherrera-Pazmiño, Linda P. Guamán

**Affiliations:** 1Escuela de Medicina, Colegio de Ciencias de la Salud Quito, Universidad San Francisco de Quito (USFQ), Quito 170901, Ecuador; 2Instituto de Microbiología, Universidad San Francisco de Quito (USFQ), Quito 170901, Ecuador; 3Facultad de Ciencia e Ingeniería en Alimentos y Biotecnología, Universidad Técnica de Ambato, Ambato 180104, Ecuador; od.lopez@uta.edu.ec (O.L.); ramirocartuchi@gmail.com (K.C.); 4Centro de Nanociencia y Nanotecnología, Universidad de Las Fuerzas Armadas ESPE, Sangolquí 171103, Ecuador; apdebut@espe.edu.ec; 5Departamento de Ciencias de la Vida y Agricultura, Universidad de las Fuerzas Armadas ESPE, Sangolquí 171103, Ecuador; 6Centro de Investigación Biomédica (CENBIO), Facultad de Ciencias de la Salud Eugenio Espejo, Universidad UTE, Quito 170527, Ecuador; arianna.mayorga@ute.edu.ec (A.M.-R.); johana.zuniga@ute.edu.ec (J.Z.-M.); rebeca.gonzalez@ute.edu.ec (R.G.-P.); antonellaviteri28@gmail.com (A.V.); ana.penaherrera@ute.edu.ec (A.B.P.-P.); 7Carrera de Ingeniería en Biotecnología de los Recursos Naturales, Universidad Politécnica Salesiana, Sede Quito, Campus El Girón, Av. 12 de Octubre N2422 y Wilson, Quito 170143, Ecuador; ecoyagoc@ups.edu.ec

**Keywords:** *Nephelium lappaceum* shell, phenolic acids, microencapsulation, spray drying, antioxidant, antimicrobial, antibiofilm, hemocompatibility, nutraceutical applications

## Abstract

Rambutan (*Nephelium lappaceum*) shell, an agro-industrial by-product, is a rich source of phenolic acids and minor anthocyanins, but its direct use is limited by instability and low bioavailability. We extracted phenolic-rich fractions and produced maltodextrin microcapsules by spray drying, then confirmed chemical entrapment and amorphization by FTIR, SEM, and XRD. The formulation showed high encapsulation efficiency and high antioxidant capacity (DPPH), selective bactericidal activity against *Pseudomonas aeruginosa* and *Burkholderia cepacia*, and strong inhibition of *Staphylococcus aureus* and *Listeria monocytogenes* biofilms, while exhibiting negligible hemolysis (<2%) across tested concentrations. Antitumor effects were moderate with low selectivity in vitro, indicating that phenolic-acid-driven redox modulation may require fractionation or delivery optimization for oncology applications. Overall, spray-dried microcapsules provided structural stability and safety while concentrating multifunctional activities relevant to food and biomedical uses. By valorizing a tropical waste stream into a bioactive, hemocompatible ingredient, this work aligns with societal goals on health and sustainable production (SDG 3 and SDG 12) and offers a scalable route to deploy underutilized phenolic resources.

## 1. Introduction

The global search for natural bioactive compounds has intensified due to growing concerns regarding synthetic additives and the prevalence of oxidative stress-related chronic diseases. Plant-derived polyphenols, including phenolic acids, flavonoids, and anthocyanins, have attracted particular attention for their multifunctional biological activities, such as antioxidant, antimicrobial, anti-inflammatory, and antitumoral effects [1,2]. However, the stability of these compounds is often compromised by environmental factors, such as pH, temperature, and light exposure, which limits their application in food and pharmaceutical formulations [3].

Agro-industrial byproducts represent an underutilized but valuable source of bioactive molecules. Fruit peels, seeds, and shells, often discarded as waste, are rich in polyphenols that can be harnessed as functional ingredients [4]. Among tropical fruits, rambutan (*N. lappaceum* L.) has received growing interest due to the high concentration of phenolic acids and anthocyanins in its shell. Previous studies have demonstrated that rambutan peel extracts possess potent antioxidant and antimicrobial activities, largely attributed to protocatechuic, syringic, and coumaric acids, along with minor contributions from anthocyanins [5,6,7]. Despite this promising bioactivity, direct use of rambutan shell extracts is hindered by poor stability, rapid degradation, and low bioavailability.

Microencapsulation has emerged as a widely adopted strategy to overcome these limitations, offering protection against environmental stressors while enabling controlled release and improved bioavailability of phenolic compounds. Among encapsulation techniques, spray drying with maltodextrin as a wall material is considered cost-effective, scalable, and highly efficient for stabilizing plant-derived extracts [8,9]. Furthermore, encapsulation not only preserves chemical integrity but can also enhance biological performance by modulating the release of active molecules during gastrointestinal transit.

In the food industry, microencapsulation of phenolics has been successfully applied to fortify beverages, dairy products, and functional snacks, extending shelf-life and masking bitterness [10,11,12,13]. In pharmaceuticals, it facilitates targeted delivery in nutraceuticals and supplements, improving therapeutic efficacy and patient compliance [14,15].

Unlike prior rambutan peel encapsulation studies that optimized spray drying or drying conditions and emphasized physicochemical/antioxidant performance in food matrices [7], this work integrates antimicrobial and antibiofilm testing against *Pseudomonas aeruginosa* and the *Burkholderia cepacia* complex—pathogens of high clinical and product-contamination concern—together with an initial hemocompatibility (hemolysis). Accordingly, our objectives were to (i) extract and spray-dry phenolic-rich fractions from *N. lappaceum* shell using maltodextrin, (ii) confirm structure/encapsulation by FTIR/SEM/XRD, and (iii) evaluate multifunctional activity (antioxidant; antimicrobial/antibiofilm; hemolysis), thereby bridging food and biomedical endpoints beyond antioxidant retention alone.

## 2. Results and Discussion

The quantification of anthocyanins in the different treatments showed values fluctuating between 22.49 ± 0.82 and 35.68 ± 0.77 mg/L. No report has been found on the yield or concentration of anthocyanins in *N. lappaceum* peel; however, since they are flavonoids, Valdez Lopez and coworker, reported concentrations of 3.60 ± 0.11 to 4.07 ± 0.18 mg flavonoids/mL, depending on the sweet or bitter *N. lappaceum* species. The data obtained in this study differ notably, as a specific flavonoid was analyzed rather than total flavonoids. Furthermore, the processing of the plant material and the extraction conditions are considered to significantly influence the final phenolic content obtained. For instance, high temperatures can reduce the bioavailability of phenols, and thermal processes generally decrease the anthocyanin content [16].

### 2.1. Anthocyanin, Phenolics, Vitamin C and Organic Acid Content

The phytochemical analysis of *N. lappaceum* shell revealed a composition dominated by phenolic acids, with *m*-coumaric acid, protocatechuic acid, and syringic acid as the most abundant constituents. Table 1. The total phenolic content (TPC, 4328.5 ± 32.9 mg/100 g DW) was markedly higher than values reported for other tropical byproducts, such as *Vaccinium* spp. or *Rubus* spp., which typically range from 10 to 30 mg GAE/g DW under similar extraction conditions [17,18].

Although the quantified anthocyanin concentration was modest (2.98 ± 0.43 mg D-ch/100 g DWM), these pigments may contribute to the overall antioxidant effect together with phenolic acids, potentially enhancing radical scavenging and membrane protection [10]. This is consistent with the potent antioxidant performance observed in the DPPH assay (IC_50_ = 19.86 µg/mL; Section 2.4), which is classified as very powerful activity [19]. The discrepancy between the relatively low anthocyanin yield and the strong antioxidant response underscores the central role of phenolic acids—particularly protocatechuic acid, which is known for its ROS-scavenging and antibacterial properties [20,21].

The predominance of *m*-coumaric acid (3514.3 ± 262.0 mg/100 g DW) is particularly relevant to the biological activities reported in later sections. This compound has been shown to modulate protein glycosylation in diabetic models [22], while syringic acid and protocatechuic acid have demonstrated complementary antitumor and anti-inflammatory effects [23,24]. Their abundance provides a biochemical rationale for the moderate antibacterial activity observed against *Pseudomonas aeruginosa* and *Burkholderia cepacia* (Section 2.5) and for the ROS-mediated cytotoxicity in antitumoral assays (Section 2.8). Nevertheless, encapsulation in a maltodextrin matrixmay restricts their immediate release, explaining the relatively high concentrations required to achieve bactericidal effects.

Vitamin C was not detected in the samples, which aligns with reports that fruit peels generally contain variable or minimal levels of ascorbic acid [25]. In contrast, high concentrations of organic acids, particularly malic (20,141.7 ± 10.18 mg/100 g DWM) and citric acids (6648.9 ± 18.9 mg/100 g DWM), were identified. These compounds can indirectly contribute to bioactivity by enhancing phenolic solubility, stabilizing anthocyanins under acidic conditions, and synergizing in antimicrobial action through disruption of microbial membranes [26]. Such interactions may partially explain the selective antibacterial activity against Gram-negative strains (Section 2.5).

Overall, these compositional findings provide a mechanistic basis for the biological outcomes observed in this study. The dominance of phenolic acids, rather than anthocyanins alone, appears to drive most of the antioxidant and antimicrobial effects, while microencapsulation ensures their stability (FTIR, Section 2.2) and safety profile (hemolysis < 2%, Section 2.8). Future studies should address the bioaccessibility and release kinetics of these phenolics under simulated gastrointestinal conditions, as previously reported for other spray-dried systems [17]. Such insights would better predict their nutraceutical potential and optimize their application in functional food and pharmaceutical formulations.

### 2.2. Structural Characterization by FTIR and Encapsulation Efficiency

The FTIR spectra of *N. lappaceum* shell extracts, before and after microencapsulation, revealed characteristic modifications that confirm the successful entrapment of anthocyanins within the maltodextrin matrix (Table 2). The most evident change was the attenuation of the broad O–H stretching band (3200–3600 cm^−1^) and the reduction in C=O stretching intensity (1700–1750 cm^−1^) in the encapsulated sample. These shifts indicate hydrogen bonding interactions between anthocyanin hydroxyl/carbonyl groups and maltodextrin, or steric shielding of reactive groups within the carbohydrate network. Similar FTIR patterns have been reported for encapsulated anthocyanins from other plant sources, confirming the stabilizing role of polysaccharide-based carriers [27,28,29].

In addition, the appearance and strengthening of polysaccharide-associated vibrations (1050–1160 cm^−1^) further confirmed the contribution of maltodextrin to the encapsulated system, while the persistence of aromatic skeletal vibrations (850–950 cm^−1^) indicated that the anthocyanin backbone remained structurally intact after spray drying. Preservation of these aromatic and glycosidic bands is particularly relevant for maintaining biological activity, as aromatic moieties are central to radical scavenging (Section 2.4) and antimicrobial interactions (Section 2.5).

Encapsulation efficiency (EE) was 84.86 ± 0.26%, consistent with a high proportion of anthocyanins retained within the maltodextrin matrix. Together with EE, microcapsules displayed amorphous, smooth morphology by SEM/XRD. These features are commonly associated with favorable dispersion and processability. In our study, however, we did not quantify phenolic degradation or compare stability versus non-encapsulated extract.

This efficiency compares favorably with values reported for other phenolic-rich extracts encapsulated by spray drying, which generally range between 70% and 85% depending on process conditions and wall materials [7,17,30]. For example, Ștefănescu et al. (2022) [31] reported efficiencies above 79% for phenolic-rich extracts microencapsulated with maltodextrin, while Boyano-Orozco et al. (2020) achieved comparable results with rambutan peel extracts [7,17,30]. The high efficiency obtained in the present study directly correlates with the FTIR data (Table 2), where attenuation of hydroxyl and carbonyl bands suggests that most reactive groups were successfully masked within the maltodextrin matrix.

The chemical evidence from FTIR is consistent with the morphological and structural results obtained by SEM and XRD (Figure 1 and Figure 2). SEM micrographs revealed predominantly spherical microcapsules with smooth surfaces, while XRD analysis confirmed the amorphous nature of the particles. This amorphization prevents crystallization of the bioactive compounds, enhances solubility, and further contributes to the stabilization of anthocyanins during storage and processing. Together, these findings validate that microencapsulation was not only efficient but also structurally effective in protecting the phenolic content identified in Section 2.1 (Table 1).

Encapsulation efficiency is a critical determinant of biological activity, as it influences both the stability and release profile of phenolic compounds. Based on morphology and phase characteristics, encapsulation could plausibly mitigate oxidative exposure; nevertheless, we did not perform time-course stability testing or side-by-side comparisons with non-encapsulated extract, and thus we cannot ascribe a protective effect. Conversely, the barrier effect of maltodextrin may also limit the immediate availability of active molecules, which could account for the relatively high concentrations required to elicit antibacterial effects (table in Section 2.5). Moreover, by masking reactive groups, encapsulation may contribute to the negligible hemolytic activity observed (<2% at 10 mg/mL, Section 2.8), underscoring the biocompatibility of the formulation.

In summary, the integration of FTIR analysis (Table 2) with encapsulation efficiency, SEM morphology (Figure 1), and XRD amorphous confirmation (Figure 2) provides a coherent characterization of the microcapsules. The spray-drying process achieved both chemical protection and structural stability, ensuring high retention of bioactive molecules and supporting their functional role in the antioxidant, antibacterial, and safety outcomes reported in the following sections.

### 2.3. Morphological and Structural Characterization of Microcapsules

The morphology of the spray-dried microcapsules was examined by SEM, revealing predominantly spherical particles with continuous and smooth surfaces (Figure 1). This morphology is characteristic of carbohydrate-based matrices such as maltodextrin, reflecting efficient atomization and drying during the encapsulation process [31]. Occasional surface wrinkling, concavities, or partially collapsed particles were also observed, which are common artifacts in spray-dried systems and typically result from rapid solvent evaporation or partial deformation under vacuum conditions [17,32]. While these irregularities do not compromise the overall morphology, they may influence flowability and bulk density, aspects relevant for industrial scale-up.

Particle size distribution analysis (Figure 1, right panel) showed an average diameter of 5.05 ± 5.76 μm, with most particles smaller than 10 μm. Similar size ranges have been reported for spray-dried anthocyanin-rich systems, where sub-10 μm particles enhance dispersibility and dissolution rates [32]. This physical property aligns well with the functional outcomes reported in subsequent assays: strong antioxidant activity (DPPH IC_50_ = 19.86 µg/mL; Section 2.4) and moderate antibacterial effects against *P. aeruginosa* and *B. cepacia* (Section 2.5).

XRD analysis confirmed the amorphous nature of the microcapsules, with a broad halo centered near 2θ ≈ 19.5° and absence of sharp crystalline reflections (Figure 2). This amorphous pattern indicates that anthocyanins were molecularly dispersed within the maltodextrin matrix during rapid drying. Amorphization has been consistently associated with enhanced solubility and stability of encapsulated phenolics [17,32]. These features correlate with the high encapsulation efficiency observed (84.86 ± 0.26%) and the negligible hemolytic activity (<2% at 10 mg/mL; Section 2.8).

Importantly, the amorphous structure also provides a rationale for the selective but moderate antibacterial activity (Section 2.5), as the smooth, non-porous surfaces may restrict rapid release of bioactive molecules, limiting immediate bactericidal effects while preserving stability. Preservation of phenolic acids such as m-coumaric, protocatechuic, and syringic acids (Section 2.1, Table 1) further supports the role of the amorphous state in maintaining bioactivity. Comparable observations have been reported for blueberry and rambutan peel encapsulates, where amorphous matrices ensured chemical stability and biological efficacy [7,32].

Interestingly, while the amorphous profile promotes enhanced antioxidant and anti-biofilm performance—as evidenced by the 81% inhibition of *S. aureus* and 93% inhibition of *L. monocytogenes* at relevant concentrations—it may also contribute to a non-specific release profile. This could explain the relatively low therapeutic indices in antitumoral assays (IC_50_ > 1000 μg/mL). The smooth, non-porous surfaces observed in SEM micrographs may further restrict targeted membrane interactions, potentially limiting the formulation’s efficacy against fungal strains such as *Candida albicans*.

In summary, the SEM and XRD data (Figure 1 and Figure 2) validate the production of spherical, amorphous microcapsules with high encapsulation efficiency and chemical stability. These structural features ensure retention of bioactivity and biocompatibility, supporting the use of *N. lappaceum* shell anthocyanins as functional ingredients in food and pharmaceutical applications.

### 2.4. Antioxidant Activity

The antioxidant capacity of microencapsulated *N. lappaceum* shell extracts were assessed using DPPH and ABTS assays, complemented by TEAC values (Table 3). The DPPH assay revealed very powerful radical scavenging activity (IC_50_ = 19.86 µg/mL), which falls below the 50 µg/mL threshold established for strong antioxidants [19,23,24,33]. This performance is remarkable considering the relatively modest anthocyanin concentration detected in the extract (Table 1), suggesting that phenolic acids—particularly m-coumaric, protocatechuic, and syringic acids—are the main contributors to the radical scavenging effect. Such compounds have been reported to act synergistically through hydrogen atom transfer and metal-chelation mechanisms, enhancing overall antioxidant potential [20,23].

The ABTS assay produced higher IC_50_ values (536.03 µg/mL), which may reflect the method’s lower sensitivity to bound or polymerized phenolics compared to the DPPH radical scavenging assay [34]. Indeed, the DPPH method is known to better capture the activity of insoluble-bound and conjugated phenolic fractions, which are abundant in rambutan shell extracts [35]. This methodological discrepancy underscores the importance of evaluating antioxidant capacity through multiple assays, as they probe different mechanistic pathways Appendix A.

The microcapsules exhibited high antioxidant activity. While encapsulation is compatible with preservation of phenolics, our study did not include a head-to-head comparison with the non-encapsulated extract; therefore, no causal inference can be made. These structural features ensure that hydroxyl-rich phenolic acids remain chemically stable, consistent with the attenuated O–H and C=O signals observed in FTIR spectra (Table 2). As a result, the antioxidant performance of encapsulated extracts was comparable to or higher than that reported for other microencapsulated tropical byproducts, such as blueberry or *Vaccinium floribundum* [7,36].

The strong antioxidant effect also correlates with the negligible hemolytic activity observed (<2% at 10 mg/mL, Section 2.8). Phenolic compounds, particularly anthocyanins and protocatechuic acid, are known to protect cellular membranes by counteracting lipid peroxidation and stabilizing redox homeostasis. This dual function—potent radical scavenging combined with biocompatibility—supports the application of *N. lappaceum* shell microcapsules in functional foods and nutraceuticals aimed at mitigating oxidative stress-related disorders.

In summary, the antioxidant activity of microencapsulated *N. lappaceum* shell is primarily driven by phenolic acids rather than anthocyanins alone. The strong DPPH scavenging activity, preserved by efficient encapsulation and amorphization, validates the potential of this agricultural byproduct as a stable source of natural antioxidants for food, pharmaceutical, and cosmetic formulations.

### 2.5. Antimicrobial Activity

The antimicrobial potential of microencapsulated *N. lappaceum* shell extract was evaluated against a panel of Gram-positive, Gram-negative, and multidrug-resistant bacterial strains, as well as four Candida species Appendix A. The microencapsulated extract demonstrated selective bactericidal activity against *Pseudomonas aeruginosa ATCC 27853* [37], *Burkholderia cepacia ATCC 25416* [38], and a multidrug-resistant clinical isolate of *P. aeruginosa*, with a minimum bactericidal concentration (MBC) of 62 mg/mL for all three strains. No antibacterial effect was detected against the remaining eleven tested strains (Table 4).

This selective activity toward Gram-negative species is noteworthy, given the intrinsic resistance mechanisms of *P. aeruginosa* and *B. cepacia*. The phenolic acids identified in Section 2.1 (Table 1), notably protocatechuic acid, *m*-coumaric acid, and syringic acid, are known to exert antibacterial effects through multiple mechanisms, including disruption of membrane integrity, chelation of metal ions, and inhibition of quorum sensing pathways [21,39,40]. Phenolic acids interfere with bacterial membranes through hydrogen bonding and hydrophobic interactions between their hydroxyl and aromatic groups and the phospholipid headgroups of the bilayer [41]. This results in membrane depolarization and leakage of cytoplasmic contents which has an adverse effect on metabolically active cells [40]. Additionally, the phenolic acids can attenuate quorum sensing by interfering with N-acyl-homoserine lactone signaling or by modulating QS-regulated gene expression, thereby reducing virulence factor production and biofilm maturation [42].

Protocatechuic acid has been reported to be potent antimicrobial agents with wide spectrum activity [43]. This compound can compromise Gram-negative envelopes and increase outer-membrane permeability, a mechanism documented for phenolic systems in *Escherichia coli* where membrane integrity and permeability rise upon exposure to protocatechuic acid–based treatments [44,45]. The catechol structure of protocatechuic acid also allows metal chelation (Fe^3+^/Cu^2+^) and generation of reactive oxygen species (ROS), leading to oxidative stress and lipid peroxidation within bacterial envelopes [21]. However, the MBC values reported here are higher than those described for pure protocatechuic acid (0.4 mg/mL against *P. aeruginosa* [21]), suggesting that bioavailability, rather than intrinsic potency, may limit activity in the encapsulated formulation. This interpretation is supported by SEM and XRD findings (Figure 1 and Figure 2), which confirmed that bioactives were entrapped within smooth, amorphous maltodextrin particles, reducing immediate release. Other notable compounds found in the extract were coumaric and syringic acids. According to the literature, the antimicrobial mechanisms of these compounds involve increased cell membrane permeabilization, leakage of cytoplasmic contents, and alterations in cellular morphology [46]. Additionally, they are potent quorum-sensing inhibitors that disrupt N-acyl homoserine lactone-mediated communication, suppress virulence genes, and prevent biofilm formation in Gram-negative bacteria [47].

No antifungal activity was observed in any *Candida* spp. at the solubility-limited maximum concentration (125 mg mL^−1^) Appendix A; higher targets (up to 500 mg mL^−1^) were unattainable due to insolubility in aqueous medium. Several factors may explain this lack of activity: (i) fungal cell walls are structurally more robust than bacterial membranes, due to their chitin and β-glucan content; (ii) the maltodextrin matrix may hinder direct interactions between phenolics and fungal cells; and (iii) the concentration tested may have been below the inhibitory threshold for Candida species. Previous studies have also shown that anthocyanins and phenolic acids display weaker activity against fungi than bacteria, unless combined with lipophilic carriers or synergistic antimicrobials [48,49,50].

Taken together, the results suggest that the antimicrobial activity of *N. lappaceum* shell microcapsules are selective, showing efficacy against certain Gram-negative bacteria but no detectable antifungal effects. This selectivity may reflect differences in cell wall permeability, release kinetics from the encapsulation matrix, and the inherent susceptibility of the target organisms. While this limits the formulation’s applicability as a broad-spectrum antimicrobial, the retained activity against clinically relevant *P. aeruginosa* strains is promising, given the pathogen’s high intrinsic resistance and clinical importance.

### 2.6. Biofilm Inhibition Activity

The microencapsulated *N. lappaceum* shell extract exhibited significant inhibitory effects on biofilm formation by *Staphylococcus aureus* and *Listeria monocytogenes*, achieving inhibition rates of 81% ± 2.87% at 100 µg/mL and 93% ± 1.41% at 1000 µg/mL, respectively (Figure 3, Appendix A). In contrast, no statistically significant inhibition was observed against *Pseudomonas aeruginosa* or *Burkholderia cepacia* at any of the concentrations tested (Figure 3). These results highlight a selective antibiofilm effect, predominantly against Gram-positive species, which are generally more susceptible to phenolic-rich extracts due to differences in cell wall structure and biofilm matrix composition [48,49].

The strong inhibition of *S. aureus* and *L. monocytogenes* biofilms can be attributed to the abundant phenolic acids identified in Section 2.1 (Table 1), particularly *m*-coumaric, protocatechuic, and syringic acids. These compounds interfere with quorum sensing, disrupt extracellular polymeric substances, and compromise bacterial adhesion [20,23,40]. Their synergistic action may explain the pronounced antibiofilm activity, despite the relatively moderate bactericidal effects observed in planktonic cultures (Section 2.5, Table 4). This distinction is clearly illustrated in Figure 3, where biofilm inhibition against Gram-positive strains reached values above 80%, while negligible effects were observed for Gram-negative pathogens.

Our study also revealed an important concentration of protocatechic acid (PCA) which has been extensively studied for their antimicrobial and antibiofilm mechanisms. For instance, a previous study showed that protocatechuic acid ethyl ester (EDHB) displays direct antibacterial activity against multiple *S. aureus* clinical isolates, with minimum inhibitory concentrations ranging from 64 to 1024 µg/mL [51]. The study also suggested that PCA derivatives may potentiate antibiotic activity by altering bacterial membrane permeability or interfering with cellular targets associated with resistance mechanisms after proving its synergy with certain antibiotics [51]. Another study analyzed the transcriptomic and molecular effects of PCA in a foodborne bacteria biofilm, *Yersinia enterocolitica*, with a biofilm inhibition efficacy of around 50.5% without affecting bacterial viability [52]. This data suggests that PCA reduced flagellar motility, cell surface hydrophobicity, extracellular polysaccharide (EPS) production, and acyl-homoserine lactone (AHL) signaling—hallmarks of quorum sensing inhibition. Gene expression analyses revealed significant downregulation of quorum-sensing regulators such as *CsrD*, *yenR*, *FlhD*, and *FlhC*, highlighting the role of PCA as a quorum-sensing modulator that interferes with the genetic control of biofilm formation [52]. These results align with our observations of reduced biofilm development, suggesting that PCA within the encapsulated *N. lappaceum* extract may similarly attenuate signaling pathways essential for biofilm maturation.

Encapsulation can also play a role in modulating antibiofilm effects. The smooth, amorphous microcapsules observed by SEM and XRD (Figure 1 and Figure 2) may have delayed phenolic release, limiting immediate activity against Gram-negative biofilms. Conversely, controlled release likely sustained bioactive availability at sub-inhibitory concentrations that interfered with quorum sensing and early adhesion stages, thereby enhancing antibiofilm performance against Gram-positive strains. This mechanism is consistent with the strong antioxidant activity retained in the encapsulated system (Section 2.4, Table 3), as oxidative stress modulation is also a recognized pathway in biofilm disruption [35].

Comparable results have been reported for microencapsulated anthocyanins from berries, where encapsulation improved stability and biofilm inhibition against *E. coli* and *Salmonella* [39]. In the present study, however, the lack of significant inhibition against *P. aeruginosa* and *B. cepacia* (Figure 3) may be attributed to the complex biofilm matrices and resistance mechanisms of these opportunistic pathogens [40].

In summary, the encapsulated *N. lappaceum* extract demonstrated selective antibiofilm activity, strongly inhibiting Gram-positive strains but not Gram-negative species. The outcomes shown in Figure 3 underscore the importance of phenolic composition, encapsulation matrix, and bacterial physiology in determining antibiofilm efficacy. Future studies should investigate synergistic strategies, such as co-encapsulation with lipophilic antimicrobials, to extend activity toward Gram-negative biofilms.

### 2.7. Antitumoral Activity

To evaluate the antitumor potential of the microencapsulated *N. lappaceum* extract, its half-maximal inhibitory concentration (IC_50_) values were determined in a panel of human tumor cell lines (HCT116, HeLa, THJ29T, and HepG2) and compared to the non-tumor fibroblast cell line NIH3T3. This comparison allowed for the calculation of the therapeutic index (TI), defined as the ratio of the IC_50_ value in non-tumor to tumor cells, serving as a measure of tumor-selective cytotoxicity. The results are detailed in Table 5. 

The microencapsulated extract of *N. lappaceum* exhibited IC_50_ values ranging from 1142 to 3571 µg/mL in the tumor cell lines. Among these, the lowest cytotoxicity in THJ29T, derived from thyroid carcinoma, suggests that this cell line may have inherent resistance mechanisms or reduced sensitivity to the treatment [53,54]. This was followed by HeLa (cervical cancer), HepG2 (hepatocellular carcinoma), and HCT116 (colorectal cancer), which exhibited progressively lower IC_50_ values, indicating increased sensitivity to the extract. Notably, microencapsulated *N. lappaceum* displayed greater cytotoxicity toward the non-tumor cells, resulting in consistently low TI values and indicating lack of tumor selectivity (TI < 1) under these conditions. The lowest TI was observed in THJ29T, while the highest TI was recorded for HCT116, although this still falls within a range of low specificity for tumor cells. 

The antitumoral activity observed for the microencapsulated *N. lappaceum* peel may be associated with its rich content of phenolic acids, particularly *m*-coumaric acid, protocatechuic acid, and syringic acid, which are recognized for their bioactive potential. These compounds play a crucial role in modulating oxidative stress and inflammatory responses by scavenging reactive oxygen species (ROS), attenuating oxidative damage, and enhancing endogenous antioxidant defenses, including superoxide dismutase, glutathione peroxidase, and catalase [22,24,55,56]. In addition to their antioxidant functions, these metabolites exhibit antitumor activity by modulating apoptotic signaling pathways, inducing cell cycle arrest, and suppressing angiogenesis and metastasis. These effects have been reported in multiple cancer models, including hepatocellular carcinoma, breast adenocarcinoma, ovarian teratocarcinoma and glioblastoma [56,57,58,59]. In this context, although low tumor selectivity was observed, this pattern likely reflects differences in intracellular accumulation, antioxidant capacity, and intrinsic resistance among the tested cell lines [60,61]. Cancer cells often upregulate efflux transporters and NRF2-mediated antioxidant systems, enhancing tolerance to oxidative stress, whereas normal fibroblasts rely on basal redox control and are more vulnerable to perturbations in redox signaling [62,63]. Although the microencapsulated extract showed modest activity, its biological relevance may extend beyond direct tumor cell killing, reflecting mechanisms such as anti-inflammatory activity, or immune modulation [64], making them attractive for use in functional foods, combinatorial therapies, and supportive care for cancer patients [65,66,67].

Beyond its bioactivity, microencapsulation aims to improve bioactive compounds’ stability, with maltodextrin acting as an effective encapsulating agent that forms a protective barrier and reduces their vulnerability to external factors. Importantly, studies have demonstrated the stability and efficacy of microencapsulated *N. lappaceum* peel in various industrial applications [7,68]. Compared to other maltodextrin-microencapsulated extracts like *Rubus glaucus* and *Vaccinium floribundum*, *N. lappaceum* exhibited markedly lower IC_50_ values despite its lower anthocyanin and phenolic content, while TI values remained similarly low [69]. These findings highlight how encapsulation can preserve or modestly enhance bioactivity, depending on the phytochemical composition and formulation strategy [70,71]. It is worth mentioning that maltodextrin is non-toxic and previous studies showed that maltodextrin alone exhibits low toxicity in tumor and non-tumor cell lines, even at concentrations well above those used in these formulations [69]. This safety profile supports its use in combination with other materials, such as ultrahigh methoxylated pectin, to create microcapsules for drug delivery [72].

In summary, the microencapsulated *N. lappaceum* shell extract demonstrates phenolic acid–mediated, dose-dependent cytotoxicity that is maintained through encapsulation but exhibits limited tumor selectivity. Future work should focus on improving selectivity through bioassay-guided fractionation and assessing sub-cytotoxic antimutagenic and redox-modulatory effects in human non-tumor cells before advancing to in vivo evaluation.

### 2.8. Hemolytic Activity

The hemolytic potential of microencapsulated anthocyanins extracted from *N. lappaceum* shell was assessed across a concentration range of 0.625 to 10 mg/mL to evaluate their cytotoxicity toward erythrocytes (Figure 4). Positive (C+, 5% Triton X-100) and negative (C−, PBS 1×) controls were included. The results revealed a consistently low hemolytic profile across all tested concentrations, with values well below the 2% threshold generally considered acceptable for biocompatible agents.

At the highest concentration (10 mg/mL), hemolysis reached only 1.9% ± 0.64%, a value indistinguishable from that observed at 2.5 mg/mL (1.9% ± 0.22%). As the concentration decreased, a modest dose-dependent decline in hemolytic activity was observed, dropping to 0.8% ± 0.09% at 1.25 mg/mL and 0.4% ± 0.27% at 0.625 mg/mL. These values underscore the negligible membrane-disruptive effect of the formulation. Standard deviations, derived through error propagation from OD measurements at 405 nm, ranged from 0.09% to 0.64%, reflecting the high reproducibility and robustness of the assay across biological replicates (n = 5). As expected, the positive control exhibited greater variability (4.99%) due to the full dynamic range of lytic response, while the negative control remained tightly consistent (0.27%). The results presented here indicate that microencapsulated *N. lappaceum* shell anthocyanins show erythrocyte compatibility and minimal cytotoxicity at physiologically relevant concentrations.

The negligible hemolytic activity can be explained by the protective effect of microencapsulation. FTIR spectra (Table 2). The hemolysis results also align with the antioxidant profile observed in Section 2.4 (Table 3). Phenolic acids such as protocatechuic acid and syringic acid are known to protect cellular membranes by scavenging reactive oxygen species and preventing lipid peroxidation [20,23]. Protocatechuic acid, in particular, has been reported to exert both antioxidative and pro-apoptotic effects depending on cellular context, while syringic acid exhibits anti-apoptotic and antioxidant activity in non-malignant systems [55].

Furthermore, the negligible hemolysis supports the selectivity of the biological responses documented throughout this study. While the extract displayed strong antibacterial (Section 2.5, Table 4), antibiofilm (Section 2.6, Figure 3), and antioxidant (Section 2.4, Table 3) activities, it did not compromise red blood cell integrity. Similar findings have been reported in phenolic-rich systems where iron and copper chelation reduces oxidative stress in membranes [73], and where synergistic interactions of bioactives can enhance antimicrobial effects without inducing cytotoxicity [74].

Future studies should focus on direct comparisons between encapsulated and non-encapsulated anthocyanins, particularly in oxidative stress models. Exploring interactions between anthocyanins and other shell-derived phenolics may yield insights into synergistic effects. Finally, clinical validation through well-designed human trials will be essential to substantiate their efficacy and safety for real-world use.

In summary, the microencapsulated *N. lappaceum* shell extract exhibited excellent hemocompatibility, with hemolysis consistently <2% (Figure 4). Together with the selective antimicrobial, antibiofilm, antioxidant, and antitumoral effects, these findings position rambutan shell microcapsules as safe and promising candidates for functional food and biomedical applications.

## 3. Materials and Methods

### 3.1. Plant Material

*N. lappaceum* shell ripe fruits were obtained in December 2024 from Ambato City Wholesale Fruit Market, Ecuador. The samples were washed, disinfected with a 100 ppm chlorine solution, and dried at 60 °C and finally stored at room temperature. The samples were ground into a fine powder using an electric laboratory blender; the fraction between 100–500 µm was collected by sieving (U.S. standard No. 35, 500 µm) [75].

### 3.2. Preparation of the Anthocyanin-Rich Extract

Ethanolic extracts were prepared according to Perez et al. (2021) with some modifications [76]. Briefly, 1.0 g of shell powder was mixed with 20 mL of acidified ethanol (96% ethanol–citric acid, 85:15, *w*/*w*)—corresponding to a solid-to-liquid ratio of 1:20 (*w*/*v*)—and subjected to microwave-assisted extraction for 5 min. The mixture was centrifuged (Hettich Rotina 380, Tuttlingen, Germany), and the supernatant was collected. The extract was then concentrated under reduced pressure using a rotary evaporator (water-bath 68 °C), then dried to constant mass in a vacuum desiccator to obtain the dry crude extract (DE).

### 3.3. Microencapsulation Process

The microencapsulation of anthocyanins was performed using maltodextrin (10–20 DE) as the carrier agent. Anthocyanin extracts with a total solids content of 30% (equivalent to 20 g of anthocyanins) were prepared at a 20:80 anthocyanins-to-maltodextrin ratio. For each extract, 280 mL of distilled water was mixed with the anthocyanin solution and combined with maltodextrin. The microencapsulation process using a laboratory-scale mini spray dryer Büchi B-290 (Flawil, St. Gallen, Switzerland). The solutions were fed into the spray dryer under controlled conditions: an inlet air temperature of 150 ± 2 °C, an outlet air temperature of 90 ± 2 °C, an internal pressure of −50 mbar, an atomizing air flow of 400–600 L/h, and a drying air flow rate of 60 m^3^/h [32]. During the spray drying process, each anthocyanin solution was atomized within the drying chamber, forming solid microspheres encapsulated in maltodextrin-based polymers. The resulting microspheres for both extracts were collected and stored in HDPE-aluminum bags at room temperature of 15–25 °C.

### 3.4. Determination of Anthocyanin, Phenolics, Vitamin C and Organic Acid Content

Analytical determinations were performed on weighed portions of DE or microcapsule powder, which were reconstituted/extracted in a defined volume solely for measurement. All results are reported per unit mass of sample. For anthocyanin quantification by the pH differential method, 20 mg of microencapsulated was weighed and mixed with 2 mL of absolute ethanol. The mixture was homogenized and stirred for 3 min in an ultrasonic bath Fisher Scientific FS60 (Fisher Scientific, Waltham, MA, USA). The supernatant was obtained by centrifugation at 14,000 rpm for 5 min at 4 °C. In a 96-well microplate, 50 µL of the microencapsulated was mixed with 200 µL of 0.025 M potassium chloride at pH 1 and 50 µL of the extract with 0.4 M sodium acetate at pH 4.5. The solutions were measured at 520 and 700 nm using a spectrophotometer with a microplate reader Cytation 5 BioTek (Agilent Technologies, Santa Clara, CA, USA) [77]. The calibration curve was constructed using a stock solution of delphinidin chloride 1 mg/mL at various dilutions. The concentration was expressed as mg delphinidin chloride per 100 g of dry weight of microencapsulated (mg D-ch/100 g DWM) [18]. Net absorbance was calculated as:A= (A_520_ − A_700_) pH1.0 − (A_520_ − A_700_) pH4.5

To extract and quantify only surface-accessible anthocyanins (SA), 100 mg of microcapsules were gently mixed with 10 mL absolute ethanol (solid/liquid = 1:100, *w*/*v*) on an orbital shaker (100 rpm, 5 min, 22–25 °C). The suspension was centrifuged (5000× *g*, 10 min), the supernatant recovered, and anthocyanins quantified by the pH-differential method as above. The short contact time and use of absolute ethanol were selected to minimize matrix swelling and diffusion from the particle interior.

The quantification of total phenolics and profiles was carried out according to the methodology of Coyago et al. [18]. For the extraction of phenolics, 20 mg of microencapsulated was weighed separately and dissolved in 1 mL of 80% methanol solution acidified with 0.1% HCl. The mixture was homogenized and sonicated for 2 min and the supernatant was recovered by centrifugation at 14,000 rpm, 4 °C for 5 min. The solid residue was re-extracted twice with 500 µL of the methanolic solution and the supernatant was recovered by centrifugation. The supernatant was stored for approximately one day until the respective analysis in freezing conditions (the solution is stable for several months) [78,79]. Total phenolics were quantified using the Folin-Cioacalteu reaction and measured in a UV-vis spectrophotometer with a microplate reader at 750 nm. Total phenolics were expressed as mg gallic acid equivalent per 100 g of dry weight (mg GAE/g DW). The phenolic profile was determined using an RRLC liquid chromatograph 1200 (Agilent Technologies, Santa Clara, CA, USA) coupled to a DAD-UV-Vis detector at a wavelength between 220 and 550 nm. In addition, a Zorbax Eclipse Plus C18 column (4.6 × 150 mm^2^, 5 µm) was used at a flow rate of 1 mL/min with a mobile phase gradient of 0.01% formic acid (solvent A) and acetonitrile (solvent B), thus, 100% at 0 min; 95% A + 5% min; 50% A +50% B at 20 min; and washing and re-equilibration of the column at 30 min. Each phenolic compound was expressed as mg/100 g of dry weight of microencapsulated (mg/100 g DWM).

Vitamin C was extracted from 20 mg of microencapsulated using 0.2 mL of a 0.2% solution of DL-homocysteine and 1.2 mL of 3% metaphosphoric acid. The resulting mixture was homogenized and sonicated in an ultrasonic bath for 3 min. Subsequently, the volume was adjusted to 2 mL with ultrapure water, and the sample was centrifuged at 14,000 rpm for 5 min at 4 °C. The supernatant was filtered using a 0.45 µm pore PVDF filter. Ascorbic acid quantification was performed by liquid chromatography (RRLC) using a diode array detector (DAD-UV-Vis) and a Zorbax Eclipse XDB-C18 column (4.6 × 50 mm^2^, 1.8 µm). The mobile phase was composed of a 90:10 mixture of 1.5% potassium phosphate monobasic solution and 1.8% n-acetyl-n,n,n-trimethylammonium bromide and pumped at 1 mL/min. Results were expressed as milligrams of ascorbic acid per 100 g dry weight of microencapsulated (mg/100 g DWM) [80].

Organic acids were extracted from 20 mg of microencapsulated with 1.5 mL of a solution of 0.02 N sulphuric acid, 0.05% metaphosphoric acid and 0.02% homocysteine. The mixture was homogenized and stirred in an ultrasonic bath for 3 min and the supernatant was recovered by centrifugation at 14,000 rpm, 4 °C for 5 min. The solids were re-extracted with 500 µL. The liquid extract was filtered through a 0.45 µm PVDF filter. Quantification was performed using RRLC, a DAD-UV-Vis detector and a YMC-Triart C18 column (150 × 4.6 mm^2^, 3, 12 nm, 400 bar) (YMC Europe GmbH, Dinslaken, Germany). The mobile phase was a 0.027% sulphuric acid solution. The concentration was expressed as milligrams per grams of dry weight of microencapsulated (mg/100 g DWM) [80].

### 3.5. Characterization by FTIR

Fourier transform infrared (FTIR) spectroscopy was employed to identify the functional groups in the microcapsules. The spectra were obtained using a Perkin Elmer Spectrum two FTIR L1600312 spectrometer (Perkin Elmer, Tokyo, Japan) with the ATR method, covering a wave number range of 4000–500 cm^−1^ at a resolution of 4 cm^−1^ across 36 scans.

### 3.6. Morphological Analysis by Scanning Electron Microscope (SEM)

The morphological study was carried out using a scanning electron microscope (SEM) (Mira 3, Tescan, Czech Republic) with electron secondary and back-scattered electron detectors. The samples were fixed onto SEM stubs using a double carbon adhesive layer, and then covered with a nearly 25 nm gold layer using a sputter coater (Q150R, Quorum, Laughton, UK). Furthermore, an X-ray diffraction (XRD) study was performed in a diffractometer (EMPYREAN, PANalytical, Almelo, The Netherlands), operating in a (θ–2θ) configuration (Bragg–Brentano geometry) equipped with an X-ray copper tube (Kα radiation λ = 1.54056 Å) at 45 kV and 40 mA. The final diffractogram was obtained by averaging 4 XRD patterns from 5° to 80°.

### 3.7. Antioxidant Capacity

Antioxidant activity was assessed by the DPPH (2,2-diphenyl-1-picrylhydrazyl) radical scavenging and by ABTS (2,2′-azino-bis (3-ethylbenzothiazoline-6-sulfonic acid)) decolorization assays. For this purpose, 3.5 mg of the microencapsulated powdered material was weighed and dissolved in 3.5 mL DMSO to a final 1 mg/mL concentration. Ascorbic acid was utilized as a standard, and the stock solution 1 mg/mL was prepared in Milli Q water. 

DPPH Microplate protocol assay was adapted from [81]. Briefly, 100 μg/mL stock extract solution and the standard were prepared as serial two-fold dilutions in methanol to a final volume of 200 µL of methanolic solutions. Afterward, the mixture was incubated in the dark at room temperature for 45 min, and the absorbance was read at 515 nm on a Cytation 5 (BioTek) plate reader.

The following formula was used to calculate the % DPPH scavenging activity:$$\%\;DPPH\;scavenging\;=\;100\;\times \;\frac{\left[\left(Abs\;Sample+\;DPPH\;\right)-\left(Abs\;Sample\;Blank\right)\right]}{\left[\left(Abs\;DPPH\;\right)-\left(Abs\;Solvent\right)\right]}$$

The amount of antioxidant activity from the microencapsulated *N. lappaceum* can be determined from the inhibitor concentration values of 50% (IC_50_). IC_50_ is the value that indicates the concentration of the sample that can inhibit radical activity by 50%. Each of the inhibition % values and known concentrations are plotted on the graph so that when y = 50 in the obtained linear equation y = mx + c of linear line equation, it will get the value of x as the value of IC_50_.

The ABTS decolorization assay was adapted from the literature [82]. Briefly, 7 mM ABTS solution was mixed with 245 mM ammonium persulfate (APS) solution to a final concentration of 2.45 APS. The solution was placed in the dark for 16 h and then diluted in water until the absorbance was 0.692 at 734 nm (ABTS radical solution). A 2 mM Trolox stock was prepared in PBS (pH = 7.2). Afterwards, it was serially diluted to obtain solutions with concentrations ranging from 12.5 to 400 μM. These were used to generate a calibration curve to determine the sample’s Trolox equivalent antioxidant capacity (TEAC) values. For this purpose, the concentrations of Trolox that produced the same percent reduction in absorbance at 734 nm for the extract and the ascorbic acid were calculated. These values were expressed as μmol TE/g.

In addition, the stock solutions of the microencapsulated *Nephelium lappaceum* extract (1.75 mg/mL) and the ascorbic acid (50 μg/mL) were prepared as serial two-fold dilutions in DMSO and water, respectively. Afterward, 10 µL of each dilution was mixed with 190 μL of ABTS radical solution (~0.7 at 734 nm) in a 96-well microtiter plate. Finally, the mixture was incubated in the dark at room temperature for 5 min, and the absorbance was read at 734 nm on a Cytation 5 (Bio Tek) plate reader.

The following formulas were used to calculate the % Decolorization: $$Decolorization\;(\%)=\left(\frac{Control\;Abs\;-Sample\;Abs}{Control\;Abs}\right)\times \;100$$

The antioxidant capacity relative to Trolox was calculated using the equation obtained for the calibration curve, and the formula:$$Trolox\;-\;eq\;\left(\frac{mg}{mg}\right)=\frac{\left(\frac{Sample\;decolorization\;(\%)-b}{a}\right)}{Sample\;concentration\;\left(\frac{mg}{mL}\right)}$$

IC_50_ values were used to express the antioxidant activity of each compound and represented the concentration that could scavenge 50% of the DPPH free radical or decolorize 50% of the ABTS radical solution. These values were determined by using GraphPad Prism 10.4.2 (GraphPad Software, Corp.) All the results are shown as a mean ± standard deviation (SD) of experiments performed in triplicate.

### 3.8. Antimicrobial Activity by Macrodilution Method

Antibacterial activity assessment was conducted using the macrodilution technique. Seven bacterial strains were selected for the antimicrobial evaluation of Gram-positives: *Staphylococcus aureus* ATCC 25923 [83], *Listeria monocytogenes* ATCC 13932 [84], *Enterococcus faecalis* ATCC 29212 [85] and Gram-negatives: *Pseudomonas aeruginosa* ATCC 27853 [37], *Burkholderia cepacea* ATCC 25416 [38], *Salmonella enterica* ATCC 14028 [86], *Escherichia coli* ATCC 25922 [87]; and seven multidrug resistant bacteria: *Klebsiella pneumoniae*, *Escherichia coli*, *Enterococcus faecalis*, *Staphylococcus epidermidis*, *Enterococcus faecium*, *Salmonella enterica* serovar Kentucky and *Pseudomonas aeruginosa.*

The minimal bactericidal concentration (MBC) was determined using the macrodilution method according to the Clinical and Laboratory Standards Institute (CSLI) guidelines [88].

This assay was performed in falcon tubes. The microencapsulated was serially diluted in distilled water; then, 1 mL of each dilution was added to 1 mL of bacterial suspension (1.5 × 10^8^ CFU/mL) to a total volume of 2 mL. The final concentration of microencapsulate in each tube ranged from 7 to 125 mg/mL. The tubes were incubated at 37 °C for 24 h with constant shaking at 180 rpm. Then, the tested tubes were centrifuged at 3000 rpm for 5 min and the pellet was resuspended in saline solution at 0.9%. 100 μL of resuspended solution was inoculated in Mueller-Hinton agar (MHA) using Drigalski loop. Plates were incubated at 37 °C for 24 h. Sterile distilled water was used as a negative control to ensure that no contamination or intrinsic antibacterial activity was present in the solvents used for extract preparation. Ampicillin (100 μg/mL), Tetracycline (10 μg/mL) and Nourseothricin (100 μg/mL) were used as a positive control. All experiments were conducted in triplicate to ensure reproducibility and reliability of the data. The results were expressed in mg/mL for each bacterial strain tested. 

### 3.9. Antifungal Activity

The antifungal activity of microencapsulated anthocyanins from *N. lappaceum* (rambutan pericarp) was evaluated against four Candida species (*C. albicans*, *C. glabrata*, *C. krusei*, and *C. tropicalis*) using a modified minimum fungicidal concentration (MFC) assay, based on the Clinical and Laboratory Standards Institute (CLSI) guidelines [89]. Each strain was grown overnight in YPD broth at 37 °C, and cell density was adjusted to an optical density (OD_650_) of 0.4 using sterile distilled water. The inocula were prepared by a 1:100 dilution of the cultures, followed by a 1:20 dilution in 2× YPD medium, ensuring a final nutrient concentration of 1× in the assay.

A stock solution of the microencapsulated anthocyanins was prepared at 250 mg mL^−1^ in sterile distilled water only; no solubilizing agents (e.g., DMSO or surfactants) were used. The maximum testable concentration was 125 mg mL^−1^ (final), which corresponded to the aqueous solubility limit of the microencapsulated extract under these conditions. For each treatment, 1 mL of this solution was mixed with 1 mL of the Candida suspension, yielding a final compound concentration of 125 mg mL^−1^ and an estimated final cell density of ~2 × 10^3^ CFU mL^−1^. The maximum testable concentration was 125 mg mL^−1^ (final), which corresponded to the aqueous solubility limit of the microencapsulated extract under these conditions. Mixtures were incubated in 2 mL volumes at 37 °C with shaking at 170 rpm for 72 h. Following incubation, cultures were centrifuged at 3000 rpm for 5 min, the supernatants discarded, and the pellets resuspended in 1 mL of sterile water. Aliquots of 100 µL were plated on YPD agar and incubated at 37 °C for 48 h to assess fungicidal activity. Higher concentrations, up to the target 500 mg mL^−1^, were not achievable due to solubility limitations of the compound in aqueous solution.

Five control conditions were included: one strain-specific control per Candida species (YPD 2× + sterile water), and a negative medium control (YPD 2× + compound at 250 mg mL^−1^ without Candida cells). 

### 3.10. Biofilm Inhibition Evaluation

The inhibitory effect of microencapsulated anthocyanins from *N. lappaceum* peel on biofilm formation of four bacterial strains (*S. aureus* ATCC 25923, *L. monocytogenes* ATCC 13932, *P. aeruginosa* ATCC 9027 [90] and *Burkholderia cepacia* ATCC 25416) was investigated using the microplate assay with crystal violet staining as described by Merritt et al. [91] with some modifications. Briefly, 1:100 bacterial dilutions were prepared in TSB+G (Tryptic Soy Broth medium supplemented with 1% Glucose) from overnight cultures and evaluated against a range of microencapsulated *N. lappaceum* (10,000 μg/mL to 1 μg/mL). 150 μL aliquots were added to 96-polystyrene plates and incubated at 37 °C in static conditions for 24 h. The medium was carefully removed using a micropipette, the wells were washed twice with PBS buffer 1X (pH 7.2) and the plate was dried inside a laboratory oven, (1 h at 60 °C). Subsequently, the biofilms were stained with 150 μL of 0.1% crystal violet for 20 min and washed three times with PBS buffer 1X (pH 7.2). Finally, 150 μL of 96% ethanol was added to each well for 30 min, and the absorbance was measured at 590 nm using a plate reader (Biotek Cytation 5). Positive controls were prepared by diluting 1:100 of the bacterial overnight culture in TSB+G.

The biofilm inhibition percentage was assessed with the following formula:$$\mathrm{Inhibitory}\;\mathrm{rate}\;(\%)\;=\;[(\mathrm{Positive}\;\mathrm{control}\;{\mathrm{OD}}_{590}\;\mathrm{nm}\;-\;\mathrm{Sample}\;{\mathrm{OD}}_{590}\;\mathrm{nm})/\mathrm{Positive}\;\mathrm{control}\;{\mathrm{OD}}_{590}\;\mathrm{nm}]\;\times 100$$

### 3.11. Antitumoral Activity

Four tumor cell lines, HeLa (human cervical carcinoma, RRID:CVCL_0030), HepG2 (hepatocellular carcinoma, ATCC: HB-8065 [92], RRID:CVCL_0027), HCT116 (human colon carcinoma, RRID: CVCL_0291), THJ29T (Thyroid carcinoma, Applied Biological Materials Inc. (abm) Cat. No. T8254, RRID:CVCL_W922), and one non-tumor cell NIH3T3 (mouse NIH/Swiss embryo fibroblasts, ATCC: CRL-1658 [93], RRID: CVCL_0594) were cultured in Dulbecco’s Modified Eagle Medium (DMEM/F12) (Corning, Corning in Manassas, VA, USA) supplemented with 10% fetal bovine serum (FBS; Eurobio, Les Ulis, France) and 1% penicillin/streptomycin (Thermo Fisher Scientific, Gibco, Miami, FL, USA). All cell lines were maintained in a humidified incubator at 37 °C with 5% CO_2_.

To evaluate the effect of microencapsulated *N. lappaceum*, HeLa, THJ29T, and NIH3T3 cell lines were seeded in 96-well plates at a density of 1 × 10^4^ cells/well in 100 μL/well of DMEM/F12. HepG2 and HCT116 were seeded at a density of 2 × 10^4^ cells/well and 1.5× 10^4^ cells/well, respectively. The following day, the medium was replaced with 100 μL of seven serial dilutions of the extract, in quadruplicate, ranging from 0.16 mg/mL to 10 mg/mL, and plates were incubated for 72 h at 37 °C with 5% CO_2_. Wells with only DMEM/F12 were used as control. The stock solution of the microencapsulation (250 mg/mL) was prepared in Ultra-pure water, stored refrigerated, and protected from light to preserve metabolite stability.

After the incubation period, cell viability was performed using the 3-(4,5-Dimethylthiazol-2-yl)-2,5- diphenyltetrazolium bromide (MTT) assay following the standard procedure provided by Sigma, St. Louis, MO, USA. Briefly, 10 μL of MTT solution (5 mg/mL) was added to each well. After 1–2 h of incubation in a humidified environment, the medium was aspirated, and 50 μL of DMSO was added to each well to dissolve the formazan crystals. The mixture was gently agitated for 1 min at 300 rpm before measuring the absorbance at 570 nm using a Cytation5 multi-mode detection system (BioTek, Winooski, VT, USA). To determine the concentration of the compound required to inhibit 50% of cell proliferation (IC_50_), dose–response curves were generated in GraphPad Prism version 10.0.0 for Windows (GraphPad Software, Boston, MA, USA, www.graphpad.com). The control group of untreated cells served as the basis for 100% cell proliferation.

### 3.12. Hemolytic Assay

The hemolytic activity of the *Nephelium lappaceum* shell extract was assessed following a previously established protocol [94]. Briefly, ten milliliters of defibrinated sheep blood were subjected to three consecutive washes with PBS 1X. Following these washes, a 1% erythrocyte suspension in PBS 1X was prepared. This erythrocyte suspension was subsequently mixed in a 1:1 ratio *N. lappaceum* shell extract at the indicated concentrations, positive controls (10% Triton X-100), or negative controls (DMSO 2.5%) in a 96-well polypropylene plate. The mixture was incubated at 37 °C for 1 h. Post incubation, the samples were centrifuged for 5 min at 1700× *g*. The supernatant was then carefully transferred to a transparent flat bottom 96-well plate for absorbance measurement at 405 nm using a Cytation5 multi-mode plate reader (BioTek). Each experiment included three technical replicates and the entire procedure was repeated three times. For each sample, the hemolysis rate was calculated according to the formula:

### 3.13. Encapsulation Efficiency (EE)

Total (TA) and surface anthocyanins (SA) were quantified by the same pH-differential method as described in Section 2.4. EE was determined from the total and surface anthocyanins, and by applying the relationship:$$EE\;(\%)=\frac{Total\;anthocyanins-Surface\;anthocyanins}{Total\;anthocyanins}\times 100$$

Each batch was analyzed in triplicate (n = 3 independent batches), each in technical duplicate; coefficients of variation were <5%. The reported value reflects mean ± SD across batches.

### 3.14. Statistical Analysis

Statistical analysis for the Biofilm Inhibition Evaluation was performed using a two-way ANOVA Dunnett test. The data of three independent biological replicates was used for the analysis. These analyses were performed and illustrated using the GraphPad Prism 10.2 software (GraphPad Software Corp, San Diego, CA, USA). The *p* values < 0.05, <0.01, and <0.001 were considered statistically significant.

## 4. Conclusions

This study demonstrates that the shell of *N. lappaceum*, an agro-industrial byproduct, is a rich source of phenolic compounds, particularly protocatechuic, syringic, and *m*-coumaric acids, which contribute to its strong bioactivity profile. Although anthocyanins were present in relatively low concentrations, their synergistic interaction with phenolic acids enhanced the extract’s functional potential.

Microencapsulation with maltodextrin proved highly efficient (84.86 ± 0.26%) and structurally effective, as evidenced by FTIR (Table 2), SEM (Figure 1), and XRD (Figure 2) analyses. The process preserved the chemical integrity of the phenolics, produced amorphous and smooth microcapsules.

*N. lappaceum* shell microcapsules retained a potent antioxidant activity; (DPPH IC_50_ = 19.86 µg/mL; Table 4); however, we did not directly compare this to the non-encapsulated extract. Selective antimicrobial properties against *P. aeruginosa* and *B. cepacia* (Table 5), and strong antibiofilm inhibition against *S. aureus* and *L. monocytogenes* (Figure 3) was displayed. Importantly, no antifungal effect was detected against *Candida* spp., indicating a selective antimicrobial spectrum. Equally relevant, the formulation exhibited excellent hemocompatibility, with hemolysis consistently below 2% (Figure 4), supporting its safety for biomedical and nutraceutical applications.

Taken together, these findings highlight the potential of *N. lappaceum* shell microcapsules as a safe, stable, and multifunctional natural ingredient with applications in food, pharmaceutical, and biomedical industries. Future research should focus on in vivo bioavailability, release kinetics, and synergistic strategies (e.g., co-encapsulation with complementary bioactives) to further expand the therapeutic and preventive applications of this underutilized byproduct.

Despite the promising bioactivity and biocompatibility demonstrated here, microencapsulated phenolics face well-known translational hurdles: establishing performance under physiologically relevant digestion, ensuring shelf-life and process robustness at scale, and linking in vitro readouts to bioaccessibility and uptake. Standardized digestion assays, paired with streamlined stability and manufacturability assessments, are needed to verify functional gains from encapsulation while preserving feasibility for real-world deployment. Future work will therefore focus on rigorous but concise digestion-stability testing and on designing validation steps that bridge laboratory efficacy with practical applications.

## Figures and Tables

**Figure 1 ijms-26-10859-f001:**
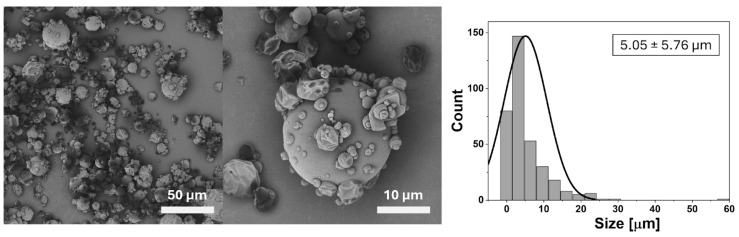
SEM micrographs and particle size distribution of anthocyanin-loaded microcapsules from *N. lappaceum* shell. Field emission scanning electron microscopy (SEM) images at low (**left**, 50 μm) and high (**center**, 10 μm) magnifications reveal predominantly spherical microcapsules with smooth surfaces, alongside occasional structural irregularities such as surface wrinkling and partial collapse—features typical of spray-dried carbohydrate matrices. The **right** panel shows the particle size distribution histogram (n = 350), displaying a unimodal profile centered around 5.05 ± 5.76 μm, with a majority of particles below 10 μm and a few outliers extending beyond 20 μm.

**Figure 2 ijms-26-10859-f002:**
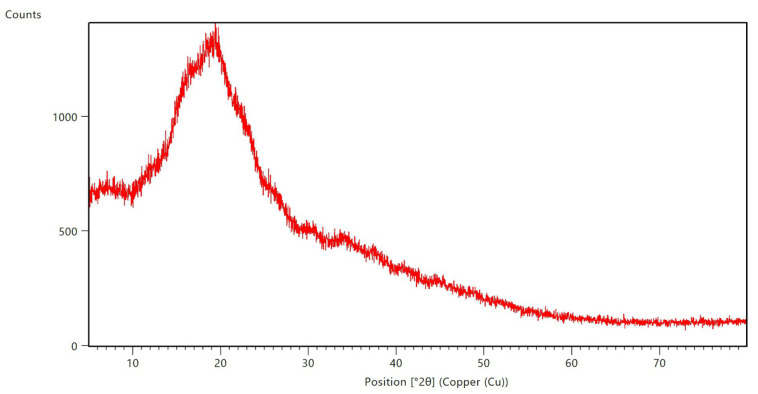
X-ray diffraction (XRD) pattern of spray-dried microcapsules containing anthocyanins from *N. lappaceum* shell. The diffractogram exhibits a broad, diffuse halo centered near 2θ = 19.5°, indicative of an amorphous structure. The absence of sharp Bragg peaks confirms that both the anthocyanin and maltodextrin components are molecularly dispersed and fully amorphized during spray drying, with no residual crystallinity detected over the scanned range (5–80° 2θ).

**Figure 3 ijms-26-10859-f003:**
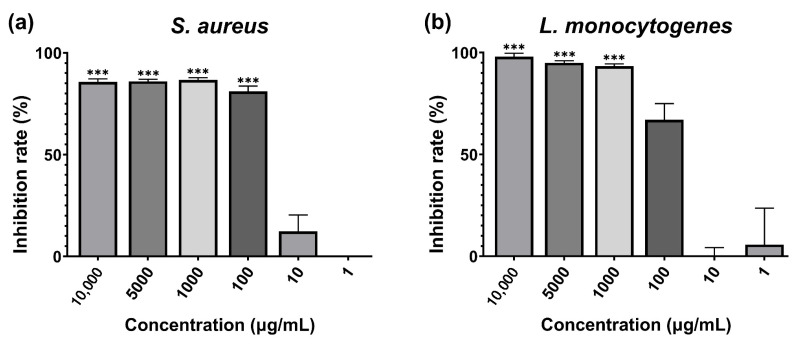
Biofilm inhibition activity of microencapsulated *N. lappaceum* shell extract against *Staphylococcus aureus*, *Listeria monocytogenes*, *Pseudomonas aeruginosa*, and *Burkholderia cepacia*. Bars represent mean ± standard deviation (n = 3). A clear concentration-dependent inhibition was observed for *S. aureus* (**a**) and *L. monocytogenes* (**b**), with maximum inhibition values of 81% ± 2.87% and 93% ± 1.41%, respectively. In contrast, no significant inhibitory effect was detected against *P. aeruginosa* or *B. cepacia*, consistent with their highly resilient biofilm matrices. Statistical differences were assessed by two-way ANOVA followed by Tukey’s test. These results indicate a selective antibiofilm effect primarily against Gram-positive strains. All the values are mean ± SD, *p*-value (***) < 0.001.

**Figure 4 ijms-26-10859-f004:**
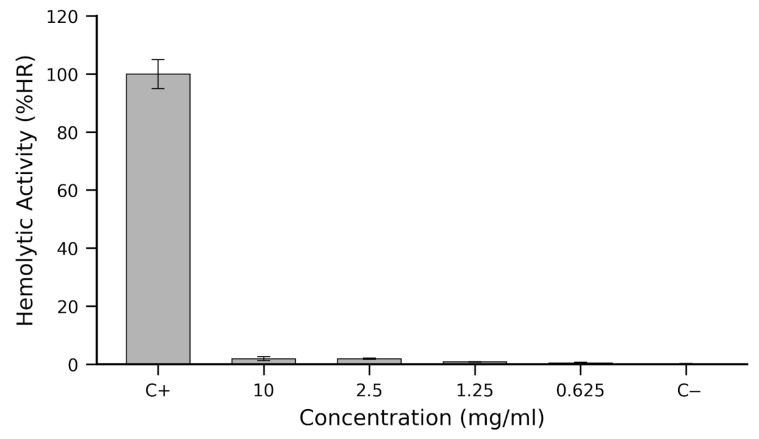
Hemolytic activity (%HR) of microencapsulated *Nephelium lappaceum* shell anthocyanins across a range of concentrations. Data represent mean ± standard deviation (SD, n = 3). C+ (positive control, 5% Triton X-100), C− (negative control, PBS 1X). The results indicate consistently low hemolytic activity at all evaluated concentrations, underscoring the minimal cytotoxicity of the microencapsulated anthocyanin formulation.

**Table 1 ijms-26-10859-t001:** Anthocyanin, phenolics, vitamin C and organic acid content.

Parameter			
Total anthocyanin (mg D-ch/100 g DW)	2.9	±	0.43
TPC (mg GAE/100 g DW)	4344.0	±	32.0
Vitamin C (mg/100 g DW)	nd		
Phenolics (mg/100 g DW)			
Gallic acid	31.0	±	1.1
Vanillic acid	15.7	±	0.7
Protocatechic acid	659.8	±	53.6
m-Coumaric acid	3514.3	±	262.0
Syringic acid	180.8	±	9.9
Kamferol acid	27.1	±	1.7
Total phenolics	4428.5	±	32.9
Organic acid (mg/100 g DW)			
Tartaric acid	2271.0	±	152.4
Malic acid	20,141.7	±	101.8
Citric acid	56,648.9	±	18.9
Total organic acid	29,061.7	±	279.4

Note: TPC, Total phenolics; D-ch, Delphinidin chloride; DW, dry weight; nd, not detectable.

**Table 2 ijms-26-10859-t002:** FTIR spectral features of microencapsulated anthocyanin extracts from *N. lappaceum* shell.

Wavenumber (cm^−1^)	Functional Group	Microencapsulated (Qualitative Intensity)
3200–3600	O-H stretching (hydroxyl)	Medium
2800–3000	C-H stretching (alkanes)	Medium
1700–1750	C=O stretching (carbonyl)	Low
1300–1350	C-H bending/aromatic ring deformation	Medium
1200–1250	C-O stretching (ethers, glycosidic bonds)	Medium
1050–1100	C-O stretching (polysaccharide-like)	Medium
1000–1050	C-O stretching (phenolic-like)	-
1140–1160	C-O-C stretching (polysaccharides)	Medium
850–950	Aromatic skeletal vibration/out-of-plane bending	Low
750–770	C-H out-of-plane bending (ring mode)	Low

**Table 3 ijms-26-10859-t003:** Microencapsulated *N. lappaceum* antioxidant activity measured by DPPH, ABTS and TEAC methods.

Compound	DPPH IC_50_ (μg/mL)	ABTS IC_50_ (μg/mL)	TEAC * (μmol TE/g)
Microencapsulated *N.lappaceum*	19.86 ± 4.7	536.03 ± 24.62	535.52 ± 24.51
Ascorbic acid	3.41 ± 1.35	68.09 ± 29.12	4678.83 ± 241.10

* TEAC based on the ABTS assay only.

**Table 4 ijms-26-10859-t004:** MBC of Microencapsulated anthocyanin extracts.

Bacterial Strain	MBC (mg/mL)
*P. aeruginosa* ATCC 27853 [37]	62
*B. cepacea* ATCC 25416 [38]	62
*P. aeruginosa* *	62

* Multidrug Resistant Bacteria.

**Table 5 ijms-26-10859-t005:** Half maximal inhibitory concentration values (IC_50_) (μg/mL) against tumor and non-tumor cell lines at 72 h and therapeutic index (TI) values. Values are expressed as mean ± standard deviation, n = 4. TI values were calculated as IC_50_ (NIH3T3)/IC_50_ (tumor cell).

	HeLa	HCT116	THJ29T	HepG2	NIH3T3
IC_50_	1853 ± 0.01	1142 ± 0.2	3571 ± 0.4	1259 ± 0.1	1092 ± 0.1
TI	0.6	1	0.3	0.9	-

## Data Availability

The original data presented in the study are openly available in FigShare at https://doi.org/10.6084/m9.figshare.30053992.

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
