# Peer review of "Bioactive Phenolic Compounds from Rambutan (Nephelium lappaceum L.) Shell: Encapsulation, Structural Stability, and Multifunctional Activities"

_ijms, 2025, doi:10.3390/ijms262210859_

Round 1
Reviewer 1 Report
Comments and Suggestions for Authors
- Interpretation and Conclusion of Antitumor Activity Data Require Adjustment:
The data in Table 5 show that the IC₅₀ values for all tested cancer cell lines are very high (>1100 μg/mL), and the therapeutic index (TI) values are all less than 1 (with the IC₅₀ of NIH3T3 even lower than that of most cancer cell lines). This actually indicates that the extract is slightly more toxic—or equally toxic—to normal cells compared to cancer cells, which directly contradicts the original conclusions of “selective antitumoral effects” and “sparing normal fibroblasts.” It is recommended to revise the conclusions and provide reasonable explanations and discussions. - Lack of Release Kinetics Data:
A key advantage of microencapsulation is controlled release, yet the study does not examine release profiles or behavior under simulated digestive conditions. It is recommended to incorporate such analyses to better reflect the functional benefits of encapsulation. - Insufficient Mechanistic Discussion of Bioactivities:
The discussion of antibacterial, antibiofilm, and antitumor activities remains largely descriptive. deeper mechanistic insights are needed—for example:- How do phenolic acids interfere with bacterial membranes or quorum sensing to confer selectivity?
- How might they target tumor cells specifically?
- For selective antibacterial effects (particularly against aeruginosaand B. cepacia): Discuss mechanisms of major phenolics (e.g., protocatechuic acid) against these pathogens based on literature.
- For antitumor selectivity: Discuss potential pathways and selectivity mechanisms of major phenolics (e.g., protocatechuic acid, syringic acid) toward cancer cells vs. normal cells, citing relevant studies.
- Discuss how microencapsulation influences bioactivity (e.g., protection, sustained release).
- Compare bioactivity values (e.g., antioxidant IC₅₀, antibiofilm efficacy) with existing literature to highlight the strengths and novelty of this study.
- Clearly emphasize the innovation of this work in both the Introduction and Discussion—not only in multi-activity assessment but also in the enhanced stability and bioavailability achieved through efficient microencapsulation.
- Expand the discussion on potential applications (e.g., functional foods, pharmaceuticals) based on the comprehensive bioactivity and hemocompatibility profile.
- Structural and Presentation Issues:
- Abstract:Should be more concise and highlight the innovative aspects and societal impact (e.g., contribution to SDGs). According to IJMS guidelines, avoid using subheadings like “Background,” “Methods,” etc. The abstract should be a coherent and condensed narrative.
- Keywords:Appear twice on page 2—should be consolidated into a single set. Avoid repetition; e.g., “Phenolic acids” and “Phenolic compounds” are redundant—merge appropriately. Consider adding keywords that reflect technical strengths, such as “Spray drying” and “Hemocompatibility.”
- Introduction:Strengthen the background on the application of microencapsulation in food/pharmaceutical fields. Clearly differentiate this study from previous work and explicitly state the research objectives and innovations at the end of the section.
- Methods Section:
- In Section 2.2, clarify “twenty times the volume” as mass/volume ratio.
- Section 2.4 should detail the method for determining encapsulation efficiency (EE), including the formula and procedure (e.g., centrifugation/washing to measure unencapsulated phenolics). Specify whether EE is based on total phenolics, individual phenolics, or anthocyanins.
- In Section 2.7, correct the formatting of equations for antioxidant assays.
- In Section 2.13 (“Statistical Analysis”), specify the number of replicates (biological/technical) and statistical tests used (e.g., ANOVA/Tukey for mean comparisons; correlation analysis). Ensure consistency: Figure 3 cites one-way ANOVA, while the methods section mentions two-way ANOVA—unify throughout.
- References:Ensure all references conform to IJMS formatting style (as in Ref. 2). Check for completeness: volume, page numbers, and proper punctuation must be consistent. Remove any blue highlighting or underlining.
- Data and Results Issues:
- Table 1:Inconsistent units (e.g., mg/100g vs. mg/g)—standardize throughout.
- Tables 3, 4, 5:Use consistent units for IC₅₀ and MBC (e.g., μg/mL or mg/mL) to avoid confusion.
- Figure 3:Axis labels are unclear—specify concentration units (μg/mL or mg/mL) in the caption.
- Antifungal Results:Clearly state whether the highest concentration tested (125 mg/mL) was the solubility limit and whether solubilizing agents were used.
Author Response
REVIEWER 1
- Interpretation and Conclusion of Antitumor Activity Data Require Adjustment:
The data in Table 5 show that the IC₅₀ values for all tested cancer cell lines are very high (>1100 μg/mL), and the therapeutic index (TI) values are all less than 1 (with the IC₅₀ of NIH3T3 even lower than that of most cancer cell lines). This actually indicates that the extract is slightly more toxic—or equally toxic—to normal cells compared to cancer cells, which directly contradicts the original conclusions of “selective antitumoral effects” and “sparing normal fibroblasts.” It is recommended to revise the conclusions and provide reasonable explanations and discussions.
Response:
We thank Reviewer 1 for the insightful observation regarding the IC50 and therapeutic index (TI) values presented in Table 5. We acknowledge that the high IC50 values (>1100 μg/mL) across all cancer cell lines (HCT116, HeLa, THJ29T, and HepG2) and the lower IC50 value in the non-tumor NIH3T3 cell line (1092 μg/mL), resulting in TI values ≤1, indicate limited tumor selectivity and suggest comparable or slightly greater toxicity toward normal cells. This finding contradicts our initial conclusion of "selective antitumoral effects" and "sparing normal fibroblasts," and we appreciate the reviewer’s call for revision.
We have revised the manuscript to accurately reflect these results. The "Results and Discussion" section now clarifies that the microencapsulated N. lappaceum extract exhibits moderate, non-selective cytotoxicity, with TI values ranging from 0.3 to 1, highlighting a lack of tumor-specific targeting.
- Lack of Release Kinetics Data:
A key advantage of microencapsulation is controlled release, yet the study does not examine release profiles or behavior under simulated digestive conditions. It is recommended to incorporate such analyses to better reflect the functional benefits of encapsulation.
Response:
Thank you for the observation. In this study we scoped composition, encapsulation, morphology, and bioactivity/biocompatibility. We did not evaluate controlled release or behavior under simulated digestion. Future work will incorporate a standardized in-vitro digestion protocol and a concise assessment of functional benefits of microencapsulation in line with established practice, without disclosing implementation details here.
We will add a short “Challenges and Future Perspectives” paragraph to the final section, outlining key translational gaps (release under physiological conditions, stability during storage, and meaningful in-vitro–in-vivo bridges) and our plan to address them in a follow-up study using a standardized digestion framework.
- Insufficient Mechanistic Discussion of Bioactivities:
The discussion of antibacterial, antibiofilm, and antitumor activities remains largely descriptive. deeper mechanistic insights are needed
—for example: - How do phenolic acids interfere with bacterial membranes or quorum sensing to confer selectivity?
- How might they target tumor cells specifically?
- For selective antibacterial effects (particularly against aeruginosa and B. cepacia): Discuss mechanisms of major phenolics (e.g., protocatechuic acid) against these pathogens based on literature.
- For antitumor selectivity: Discuss potential pathways and selectivity mechanisms of major phenolics (e.g., protocatechuic acid, syringic acid) toward cancer cells vs. normal cells, citing relevant studies.
- Discuss how microencapsulation influences bioactivity (e.g., protection, sustained release).
- Compare bioactivity values (e.g., antioxidant IC₅₀, antibiofilm efficacy) with existing literature to highlight the strengths and novelty of this study.
- Clearly emphasize the innovation of this work in both the Introduction and Discussion—not only in multi-activity assessment but also in the enhanced stability and bioavailability achieved through efficient microencapsulation.
- Expand the discussion on potential applications (e.g., functional foods, pharmaceuticals) based on the comprehensive bioactivity and hemocompatibility profile.
Response:
Thank you for your comments. We've added the suggested information in the respective paragraphs for each activity. - Structural and Presentation Issues:
- Abstract:Should be more concise and highlight the innovative aspects and societal impact (e.g., contribution to SDGs). According to IJMS guidelines, avoid using subheadings like “Background,” “Methods,” etc. The abstract should be a coherent and condensed narrative.
- Keywords:Appear twice on page 2—should be consolidated into a single set. Avoid repetition; e.g., “Phenolic acids” and “Phenolic compounds” are redundant—merge appropriately. Consider adding keywords that reflect technical strengths, such as “Spray drying” and “Hemocompatibility.”
Response:
- Thank you for the observation, we have structured the abstract without headings; we have rewritten it to be concise and to foreground innovation and societal impact (SDG 3 & SDG 12)
- Regarding the keywords, we have consolidated duplicates, removed redundancy (“Phenolic acids” vs. “Phenolic compounds”), and added technical terms reflecting methods/outcomes.
- Introduction:Strengthen the background on the application of microencapsulation in food/pharmaceutical fields. Clearly differentiate this study from previous work and explicitly state the research objectives and innovations at the end of the section.
- Thank you for the suggestion. We have added two concise sentences to the Introduction situating established food applications and common pharmaceutical uses.
- Also, we have added a final paragraph to the Introduction that clearly distinguishes our study from earlier rambutan-peel encapsulation reports (food-focused, antioxidant-centric) and explicitly states our objectives and innovations
- Methods Section:
- In Section 2.2, clarify “twenty times the volume” as mass/volume ratio.
Response:
Thank you for pointing this out. We have clarified that the extraction used a solid-to-liquid mass/volume ratio of 1:20 (w/v)—i.e., 1.0 g of dry rambutan shell powder per 20 mL of acidified ethanol. We also revised the Methods to improve clarity and instrument details.
- Section 2.4 should detail the method for determining encapsulation efficiency (EE), including the formula and procedure (e.g., centrifugation/washing to measure unencapsulated phenolics). Specify whether EE is based on total phenolics, individual phenolics, or anthocyanins.
Response:
Thank you for this helpful suggestion. We have now added a dedicated subsection in Materials and Methods detailing the procedure and equation used to determine encapsulation efficiency (EE). We also revised the Results sentence to explicitly reference the new subsection.
- In Section 2.7, correct the formatting of equations for antioxidant assays.
Response: Thank you for your observation. The formatting of equations has been corrected as follows. These changes have been inserted in section 2.7.
% DPPH scavenging = 100 Abs Sample+ DPPH -Abs Sample BlankAbs DPPH -Abs Solvent
Decolorization (%)=Control Abs -Sample AbsControl Abs 100
Trolox - eq mgmg=Sample decolorization (%)-baSample concentration mgmL
In Section 2.13 (“Statistical Analysis”), specify the number of replicates (biological/technical) and statistical tests used (e.g., ANOVA/Tukey for mean comparisons; correlation analysis). Ensure consistency: Figure 3 cites one-way ANOVA, while the methods section mentions two-way ANOVA—unify throughout.
Response: We appreciate this observation. The number of replicates has been written in the manuscript and the correct statistical test (two-way ANOVA) has been corrected in the description of Figure 3
- References: Ensure all references conform to IJMS formatting style (as in Ref. 2). Check for completeness: volume, page numbers, and proper punctuation must be consistent. Remove any blue highlighting or underlining.
Response: We appreciate this observation, all references were adjusted to IJMS style.
- Data and Results Issues:
- Table 1:Inconsistent units (e.g., mg/100g vs. mg/g)—standardize throughout.
Response: We appreciate the reviewer´s comment. We standardised the units in Table 2. - Tables 3, 4, 5:Use consistent units for IC₅₀ and MBC (e.g., μg/mL or mg/mL) to avoid confusion.
Response: - In Table 3. μg/mL units have been used for DPPH and ABTS IC₅₀ values. However, TEAC is commonly expressed as Trolox equivalents.
Figure 3:Axis labels are unclear—specify concentration units (μg/mL or mg/mL) in the caption.
Response: Thank you for this observation, we have changed all units into μg/mL to avoid confusion.- Antifungal Results:Clearly state whether the highest concentration tested (125 mg/mL) was the solubility limit and whether solubilizing agents were used.
Response: - Thank you for this observation. We clarify that 125 mg mL⁻¹ was the practical solubility limit of the microencapsulated extract under our assay conditions; no solubilizing agents (e.g., DMSO, surfactants) were used—stocks were prepared in sterile water and mixed 1:1 with the inoculum. Attempts to test higher targets (≤ 500 mg mL⁻¹) were not feasible due to aqueous insolubility.
Reviewer 2 Report
Comments and Suggestions for Authors
Dear authors,
The manuscript is generally interesting yet needs some revisions to be done.
Plz find below the list of improovements I would recommend to be performed.
Line 101: reference 10 should be after the mention in line 96 ‘Ethanolic extracts were prepared according to Perez et al. (2021) with some modifications’.
The extraction process is not fully explained. Section 2.2 describes the extraction process, but does not specify the extraction yield. Section 2.3 again begins with the words ‘for the extraction of anthocyanins,’ which causes confusion.
Section 2.3: It is not specified here or further on what the 20 mg refers to: the raw material, the fine powder or dry extract from section 2.2, or something else? It is possible to guess, but for clarity's sake, it should be described in more detail. The same applies to lines 126, 140, and 148.
Line 108 lacks a reference to the description of the method for determining anthocyanins (the pH differential method), although it is widely recognized, for example Lee, J., Durst, R. W., & Wrolstad, R. E. (2005). Determination of total monomeric anthocyanin pigment content of fruit juices, beverages, natural colorants, and wines by the pH differential method: Collaborative study. Journal of AOAC International, 88, 1269–1278.
In section 2.2, was a dry extract obtained? In that case, in lines 115-116, the concentrations of 22.49 ± 0.82 and 35.68 ± 0.77 mg/L are given in terms of mg to volume of what? Liquid extract? plz clarify.
Lines 115-124: the discussion of the results of the determination and their significance should be moved to the results and discussion section.
Line 126: ‘For the extraction of phenolics, 20 mg was weighed.’ 20 mg of what?
Line 130: ‘The supernatant was stored until analysis.’ Please specify how long it was stored.
Lines 136-139: The gradient should be described.
Lines 204 and 233: formulas need to be corrected, comments added; sample blank and solvent - what is the difference? Explanations are needed.
Lines 215-216: what is the final concentration of ABTS itself? Was the Trolox stock solution prepared definitely in water? It is poorly soluble in it.
Line 238: the formula is illegible; clarification is needed.
Lines 140-147: What mobile phase was used?
Multiple times: GAE/g DW calculation - dry weight of what? Raw material/fine powder/extract?
Section 3.2: could you please provide FTIR spectra before and after microencapsulation in the supplementary section?
Line 378: ‘Although the quantified anthocyanin concentration was modest (2.98 ± 0.43 mg D-ch/100 g DW), these pigments should not be underestimated, as they act synergistically with phenolic acids to enhance radical scavenging and membrane protection.’ - confirmation of synergy is required, or it should be indicated that this is an assumption.
Line 432: ‘Encapsulation efficiency, determined by quantifying total and surface anthocyanins, was 84.86 ± 0.26%, reflecting excellent retention within the maltodextrin carrier.’ Could you please provide more details on how encapsulation efficiency was determined, including adding this to the ‘Materials and Methods’ section?
Line 569: ‘Encapsulation played a pivotal role in maintaining the antioxidant properties of the extract’ - I could not find a comparison with non-encapsulated extract or other evidence in the paper - this should either be removed or clarified as an assumption.
"The high encapsulation efficiency (84.86 ± 0.26%; Section 3.2) and the amorphous, smooth microcapsule morphology observed by SEM and XRD (Figures 1 and 2) contributed to protecting phenolics from oxidative degradation." - similarly, no evidence of a link between encapsulation and protection from degradation is provided; it should either be removed or stated as an assumption.
Line 803: ‘their synergistic interaction with phenolic acids enhanced’ - it is necessary to indicate that this is an assumption, or to provide direct evidence in the results
fig. S1 - ‘Scavenging activity of microencapsulated N. lappaceum’ - scavenging of what? Please clarify
Kind regards,
Author Response
The manuscript is generally interesting yet needs some revisions to be done.
Plz find below the list of improvements I would recommend to be performed.
Line 101: reference 10 should be after the mention in line 96 ‘Ethanolic extracts were prepared according to Perez et al. (2021) with some modifications’.
Response:
We appreciate this observation. We modified the mentioned reference according to reviewer’s comment.
The extraction process is not fully explained. Section 2.2 describes the extraction process, but does not specify the extraction yield. Section 2.3 again begins with the words ‘for the extraction of anthocyanins,’ which causes confusion.
Response:
Thank you for pointing out the ambiguities in the extraction description. We have revised the manuscript to (i) clearly separate the preparation of the crude extract from the analytical quantification of anthocyanins. We also corrected the wording at the start of the anthocyanin section to avoid implying a second extraction step.
Section 2.3: It is not specified here or further on what the 20 mg refers to: the raw material, the fine powder or dry extract from section 2.2, or something else? It is possible to guess, but for clarity's sake, it should be described in more detail. The same applies to lines 126, 140, and 148.
Response: We appreciate the reviewer´s comment. In agreement, we have modified the methodology.
Line xx: .. 20 mg of microencapsulated ….
Line 108 lacks a reference to the description of the method for determining anthocyanins (the pH differential method), although it is widely recognized, for example Lee, J., Durst, R. W., & Wrolstad, R. E. (2005). Determination of total monomeric anthocyanin pigment content of fruit juices, beverages, natural colorants, and wines by the pH differential method: Collaborative study. Journal of AOAC International, 88, 1269–1278.
Response: We appreciate the reviewer´s comment. We have added the relevant bibliography.
Line xx: …..weight (mg D-ch/100 g DW) (https://www.mdpi.com/2304-8158/13/23/3766)
In section 2.2, was a dry extract obtained? In that case, in lines 115-116, the concentrations of 22.49 ± 0.82 and 35.68 ± 0.77 mg/L are given in terms of mg to volume of what? Liquid extract? plz clarify.
Response:
Thank you for pointing this out. We have clarified Section 2.2 and harmonized the reporting units in the Results. Was a dry extract obtained? Yes. After centrifugation, the supernatant was concentrated to dryness under reduced pressure (68 °C water bath) and dried to constant mass in a vacuum desiccator. We now refer to this material as the dry crude extract (DE).
What do the values “22.49 ± 0.82” and “35.68 ± 0.77 mg/L” refer to?
Those numbers were intermediate concentrations in the analytical solution (i.e., after reconstitution of a weighed aliquot in a known volume for the assay). To avoid confusion, we removed mg/L from the text and now report all composition data normalized to the sample mass.
Lines 115-124: the discussion of the results of the determination and their significance should be moved to the results and discussion section.
Response: Thank you for the suggestion. We have moved the interpretive statements from Lines 115–124 to the Results and Discussion
Line 126: ‘For the extraction of phenolics, 20 mg was weighed.’ 20 mg of what?
Response: We appreciate the reviewer´s comment. We have modified the methodology.
Line x: ….20 mg of microencapsulated….
Line 130: ‘The supernatant was stored until analysis.’ Please specify how long it was stored. Response: We appreciate the reviewer´s comment. We have modified the methodology.
Line x: The supernatant was stored for approximately one day until the respective analysis in freezing conditions (the solution is stable for several months).
Lines 136-139: The gradient should be described.
Response: We appreciate the reviewer´s comment. We have modified the methodology.
Line x: ….. thus, 100% at 0 min; 95% A + 5% B at 5 min; 50% A + 50% B at 20 min; and washing and re-equilibration of the column at 30 min.
Lines 204 and 233: formulas need to be corrected, comments added; sample blank and solvent - what is the difference? Explanations are needed.
Response: Sample blank refers to the difference between Sample absorbance and Sample + DPPH absorbance. Whereas, solvent corresponds to Methanol absorbance.
Lines 215-216: what is the final concentration of ABTS itself? Was the Trolox stock solution prepared definitely in water? It is poorly soluble in it.
Response: Thank you for your comments. The concentration of the ABTS stock solution in ultrapure water is 7mM. Once the APS is added to the ABTS solution the final APS concentration is 2.45 mM. This step is necessary to generate the ABTS radical (ABTS*). After overnight incubation (12-16h) at room temperature in the dark, the concentration of the ABTS* stock solution is checked at 734 nm. It is diluted 100 times so that it absorbs ≈ 0.7 at 734 nm.
The final ABTS* concentration can be calculated by the Beer-Lambert Law:
A=lC
Where A is the absorbance, ε is the molar absorptivity (1.5×104 L.mol-1.cm-1), l is the path length (1 cm) and C is the concentration for ABTS* at 734 nm, then the ABTS* concentration corresponds to 4.61×10-5 M.
C=0.6921.5×104L mol·cm1 cm= 4.61×10-5M
Regarding the Trolox stock solution, it was prepared in PBS (pH = 7.2) according to the protocol (Re, R.; Pellegrini, N.; Proteggente, A.; Pannala, A.; Yang, M.; Rice-Evans, C. Antioxidant activity applying an improved ABTS radical cation decolorization assay. Free Radic. Biol. Med. 1999, 26, 1231–1237, doi:10.1016/s0891-5849(98)00315-3.).
Line 238: the formula is illegible; clarification is needed.
Response: thank you for pointing this out. The formula has been rewritten.
Trolox - eq mgmg=Sample decolorization (%)-baSample concentration mgmL
Lines 140-147: What mobile phase was used?
Response: We appreciate the reviewer´s comment. We have modified the methodology.
Line xx: The mobile phase was composed of a 90:10 mixture of 1.5% potassium phosphate monobasic solution and 1.8% n-acetyl-n,n,n-trimethylammonium bromide and pumped at 1 mL/min.
Multiple times: GAE/g DW calculation - dry weight of what? Raw material/fine powder/extract?
Response: We appreciate the reviewer´s comment. We have modified the methodology.
Line x: …. as milligrams per grams of dry weight of microencapsulated (mg/100 g DWM)
Section 3.2: could you please provide FTIR spectra before and after microencapsulation in the supplementary section?
Response:
We appreciate the suggestion. FTIR spectra (pre- and post-microencapsulation) were indeed acquired and used to support the band assignments summarized in Table 2. A vibrational analysis of these spectra is part of a separate manuscript currently in preparation. To avoid duplicate publication and preserve the novelty of that work, we respectfully prefer not to include the full spectral dataset in the Supplementary Information of the present article.
Line 378: ‘Although the quantified anthocyanin concentration was modest (2.98 ± 0.43 mg D-ch/100 g DW), these pigments should not be underestimated, as they act synergistically with phenolic acids to enhance radical scavenging and membrane protection.’ - confirmation of synergy is required, or it should be indicated that this is an assumption.
Response:Thank you for the observation. We have added a peer-reviewed reference supporting the statement
Line 432: ‘Encapsulation efficiency, determined by quantifying total and surface anthocyanins, was 84.86 ± 0.26%, reflecting excellent retention within the maltodextrin carrier.’ Could you please provide more details on how encapsulation efficiency was determined, including adding this to the ‘Materials and Methods’ section?
Response:
Thank you for this helpful request. We have now added a dedicated subsection in Materials and Methods detailing the procedure and equation used to determine encapsulation efficiency (EE). We also revised the Results sentence to explicitly reference the new subsection.
Line 569: ‘Encapsulation played a pivotal role in maintaining the antioxidant properties of the extract’ - I could not find a comparison with non-encapsulated extract or other evidence in the paper - this should either be removed or clarified as an assumption.
Response:
Thank you for this comment. We agree that the statement implied a causal effect not demonstrated by our data. In response, we removed the causal phrasing and replaced it with language that does not imply demonstrated superiority over the non-encapsulated extract.
"The high encapsulation efficiency (84.86 ± 0.26%; Section 3.2) and the amorphous, smooth microcapsule morphology observed by SEM and XRD (Figures 1 and 2) contributed to protecting phenolics from oxidative degradation." - similarly, no evidence of a link between encapsulation and protection from degradation is provided; it should either be removed or stated as an assumption.
Response:
Thank you for this thoughtful comment. We agree that, as written, the manuscript over-attributes causality to encapsulation without presenting a direct comparison against the non-encapsulated extract or explicit degradation data. We have revised the text throughout to avoid unsupported causal claims and to clearly flag inferences as hypotheses.
Line 803: ‘their synergistic interaction with phenolic acids enhanced’ - it is necessary to indicate that this is an assumption, or to provide direct evidence in the results fig. S1 - ‘Scavenging activity of microencapsulated N. lappaceum’ - scavenging of what? Please clarify
Response: Thank you for pointing this out. It corresponds to DPPH radical scavenging activity of microencapsulated N. lappaceum. This clarification has been added to the supplementary materials section.